# Pre-Hospital Emergency Medical Services Utilization Amid COVID-19 in 2020: Descriptive Study Based on Routinely Collected Dispatch Data in Bavaria, Germany

**DOI:** 10.3390/healthcare11141983

**Published:** 2023-07-08

**Authors:** Kathrin Hegenberg, Alexander Althammer, Christian Gehring, Stephan Prueckner, Heiko Trentzsch

**Affiliations:** Institut für Notfallmedizin und Medizinmanagement, Klinikum der Universität München, Ludwigs-Maximilians-Universität München, Schillerstr. 53, 80336 Munich, Germany; alexander.althammer@outlook.com (A.A.); christian.gehring@med.uni-muenchen.de (C.G.); stephan.prueckner@med.uni-muenchen.de (S.P.); heiko.trentzsch@med.uni-muenchen.de (H.T.)

**Keywords:** emergency medical services, ambulance, utilization, pre-hospital, COVID-19

## Abstract

Background and Importance: The COVID-19 pandemic affected the utilization of health care services and posed organizational challenges. While many previous studies focused on the misuse of pre-hospital EMS for low-urgency health problems, the pandemic has put more emphasis on the avoidance of medically necessary calls. Objective: To compare the utilization of pre-hospital emergency medical services before and after specific pandemic periods. Design, setting and participants: This was a retrospective, descriptive analysis of routine data from 26 dispatch centers in Bavaria, Germany. Outcomes measure and analysis: We investigated the number of emergencies per 100,000 population, as well as the relative change in the emergency rates and transport rates in 2020, compared to the two previous years. Boxplots showed the distributions across the Bavarian districts per calendar week. The mean rates and standard deviations as well as the relative changes were presented for the specific periods. A paired samples *t*-test was used to compare the rates. Main results: Compared to the average of the two previous years, the emergency rates in 2020 were lower in 35 out of 52 calendar weeks. The strongest reductions were observed during the first wave, where the average emergency rate declined by 12.9% (SD 6.8, *p* < 0.001). There was no statistically significant difference in the overall emergency rate during the summer holidays. Lower transport rates were observed throughout the year, especially during the first wave. Conclusions: Utilization of pre-hospital emergency medical services decreased in 2020, especially during the periods with strict measures. This could be due to the lower morbidity from the behavioral changes during the pandemic, but also to the avoidance of medical services for both less urgent and severe conditions. While a reduction in unnecessary care would be beneficial, patients must be encouraged to seek necessary urgent care, even during a pandemic.

## 1. Introduction

The rising utilization of pre-hospital emergency medical services (EMS) has been a trend in Bavaria over many years [1]. This came with concerns about overwhelming the EMS system, even more so in a global pandemic. To preserve the healthcare systems, non-pharmaceutical interventions expected to contain and limit the spread and to mitigate the impact of SARS-CoV-2 were implemented in many countries. These strategies also affected healthcare and EMS utilization.

While previous studies regarding pre-hospital EMS have often revolved around the misuse for primary care and low-urgency health problems [2], the pandemic put the avoidance of medically necessary calls in the spotlight. A decline in presentations to emergency departments (ED) was detected early in the pandemic [3,4]. To date, articles regarding the utilization of pre-hospital EMS in Germany during the pandemic also reported a decline in EMS use [5,6,7,8]. These studies were limited to the observation of smaller regions and considered only a short time period during the initial lock-down in early 2020.

The aim of this study was, therefore, to investigate the use of pre-hospital EMS during the COVID-19 pandemic throughout the entire year of 2020 on a superregional level, and to draw comparisons with previous years. To provide information for different periods of the COVID-19 pandemic and to reveal the effects of counter measures, we investigated emergency and transport rates and identified differences between years, specific periods, emergencies with and without the dispatch of emergency physicians, urban and rural districts as well as dispatch keyword categories. 

## 2. Methods

### 2.1. Design and Setting

This was a retrospective observational study based on routinely collected ambulance dispatch data from Bavaria, Germany, which has a population of approximately 13 million. Pre-hospital medical services are accessed via a national emergency telephone number (112). Regional dispatch centers coordinate emergency and non-emergency ground and air ambulances as well as fire brigades. Based on a keyword-based dispatch manual, dispatchers decide on the type and number of units to be dispatched to the scene. The EMS system is two-tiered: The standard response to an emergency is an ambulance staffed with paramedics. Physicians are dispatched according to an indication catalogue when vital signs are suspected to be unstable, when the condition implicates a high probability of need for invasive intervention, if the paramedic staff on the scene requests a physician or at the discretion of the dispatcher e.g., if a prolonged response time interval for the paramedic staffed ambulance is expected.

The first confirmed COVID-19 case in Germany was reported on 27 January 2020 near Munich, Bavaria. The first cluster of cases was fully contained, yet cases began to rise in March. To identify the relevant periods during the SARS-CoV-2 pandemic in Bavaria in 2020, we screened official regulation documents (Bayerische Infektionsschutzmaßnahmenverordnung (BayIfSM)) and extracted the relevant dates. The periods we chose to analyze were mainly characterized by rigorous contact restrictions, closed restaurants and shops as well as distance learning or restricted access to schools and daycare centers. We also analysed the utilization during Bavarian summer holidays, when most restrictions were lifted. Detailed information of the measures in place during the respective periods is available in Appendix A.

### 2.2. Data Collection and Variables

Electronic records are automatically generated for each dispatch. Records of all 26 Bavarian dispatch centers are transferred to a central relational database each month. Following the integration, data were screened for suspected data abnormalities and conspicuous data points were investigated. If multiple dispatches corresponded to the same event, they could be assigned to this event and be analyzed as one single emergency event. Information about the time and location of an emergency, physician support, subsequent transportation to hospital and dispatch keyword were extracted if emergencies met the following criteria: emergencies with paramedic- or physician-staffed emergency vehicle within Bavaria between the years 2018 and 2020 documented by a Bavarian dispatch center. For the analysis of transport rates, the sample was restricted to vehicles that were equipped to transport patients. Events with dispatch of more than five vehicles were also excluded. We assumed that a patient was transported when timestamps either indicated the departure from the scene, the arrival at the hospital or if a hospital was documented as the transport destination.

Since dispatch keywords are not fully standardized, 638 different keywords were condensed and classified into 32 categories. Analyses of the 10 most frequent emergency dispatch keyword categories were displayed. Ninety-three percent of all emergencies could be assigned to one of these categories (see Appendix A for explanation of the 10 most common dispatch keywords).

Based on the emergency location, we assigned every emergency to one of the 96 Bavarian districts. According to a classification by the “Federal Institute for Research on Building, Urban Affairs and Spatial Development” (BBSR) [9], each district was allocated to a level of rurality. We distinguished four levels: large city (*n* = 8), urban area (*n* = 20), predominantly rural district with urban agglomerations (*n* = 33) and sparsely populated rural district (*n* = 35).

To calculate the per capita emergency rates based on the Bavarian population, we obtained data about the estimated monthly population in each Bavarian district from the Bavarian State Office for Statistics [10]. As these data were not available for December 2020, the population as of November 2020 was used. The number of inhabitants per calendar week within a month was assumed to be constant. If a calendar week fell in two different months, we assumed the number of inhabitants for the month at the beginning of the week.

Aggregated mobility data for Bavaria were downloaded from the website of the Federal Statistical Office [11].

We downloaded the data of confirmed COVID-19 cases per 100,000 inhabitants from the data platform of the Robert Koch Institute, Berlin [12].

### 2.3. Analysis

Utilization of EMS was reported as the number of emergencies per 100,000 population (emergency rate). The rate referred to either emergencies per 100,000 population per calendar week, or to emergencies per 100,000 population per pandemic period (first lockdown, lockdown “light” period, second lockdown and holiday period). Calendar weeks and periods were compared with the corresponding weeks or periods in previous years. For this purpose, the rates of 2018 and 2019 were averaged. For each Bavarian district, we calculated the relative changes in emergency rates (%) as a difference between the year 2020 and the average of the two previous years.

The distribution of the relative change in emergency rates across Bavarian districts per calendar week, distribution of the 7-day incidence and distribution of transport rates were presented as boxplots. The daily change in mobility was pre-calculated by the Federal Statistical Office and was summarized to obtain the average change per calendar week. Outliers (data points more than 1.5 interquartile ranges below the 1st quartile or above the 3^rd^ quartile) were not displayed.

For each pandemic period in 2020, data were presented as the mean rates ± standard deviation (SD) as well as the mean relative change ± SD. We use a paired samples *t*-test with Bonferroni correction to compare the mean emergency rates of Bavarian districts between 2018/2019 and 2020. Relative changes in transport rates per period were presented as boxplots without outliers. Data analysis was conducted using R statistical software (version 4.1.3).

## 3. Results

The total sample included 3,150,756 emergencies, 1,051,635 in 2018, 1,074,577 in 2019 and 1,024,544 in 2020. Throughout the observed period the overall minimum emergency rate per calendar week and district was 36, the maximum rate was 422. Emergency and transport rates per calendar week in 2020 as well as the average rates in 2018 and 2019 are shown in Appendix A.

### 3.1. Utilization Per Calendar Week

The relevant periods during the SARS-CoV-2 pandemic in 2020 are marked in Figure 1a. They included two periods with strict measures, including school and day-care closures (first lockdown, as well as the beginning of the second lockdown in 2020) and a period with restrictions on the number of social contacts, closed restaurants and shops, but with open schools and day care (lockdown “light” period) as well as the Bavarian summer holidays. The graph also shows the first COVID-19 case in Germany, near Munich, Bavaria, and marks the two periods where the government declared a state of emergency in Bavaria.

Figure 1b shows the distribution of the 7-day incidence in Bavarian districts in the calendar weeks 1-52. The start of the first and second wave around calendar week 10 and 37, respectively, were well observable. The highest average 7-day incidences were observed during the start of the second wave at the end of the year, with a maximum 7-day incidence of 647.2 per 100,000 in the Bavarian district Regen in calendar week 50. 

Figure 1c shows that declines in mobility preceded the restrictions, and were more pronounced during the first wave. The strongest decline in mobility (−45.5%) was observed in the calendar week starting on the 23rd of March.

The strongest median reduction in the emergency rates was during the first wave in calendar week 17 (starting 27 April) (−17.8%), whereas the highest median increase in the emergency rates was observed in calendar week 32 (starting 10 August) (+10.0%), during the summer holidays (Figure 1d). Overall, the emergency rates fell shortly before the restrictions were implemented in spring, and remained at lower levels than in the two previous years until the end of July. Another decline occurred from the beginning of November. Throughout the year, compared to the average of the two previous years, the emergency rate in 2020 was lower in 35 out of 52 calendar weeks.

The median transport rates in every calendar week in 2020 remained below the average median transport rates of 2018 and 2019 (Figure 1e). In 2018/2019, the median transport rates ranged between 78 and 80% whereas this range was 70 and 80% in 2020 (see Appendix A). Low transport rates were especially observed during the first wave.

### 3.2. Utilization during Specified Periods 

During the first lockdown, the overall emergency rate declined by 12.9% (SD 6.8, *p* < 0.001) (Table 1). The reduction was almost the same for emergencies with and without emergency physician support. Most emergencies were observed in large cities, where the reduction in the rates was also most prominent (−17.2% (SD 6.6, *p* < 0.001)). The most frequent reason for dispatch in 2020, as well as in the previous years, was cardiovascular complaints (2018/2019: 249.7 (SD 48.1), 2020: 228.4 (SD 40.7)). During the first lockdown, cardiovascular emergencies declined by 7.8% (SD 10.1, *p* < 0.001). A strong statistically significant decline of about 30% was observed for dispatch keywords indicating pediatric emergencies and traffic accidents, whereas the emergency rates for conditions indicating respiratory problems increased by 10.1% (SD 20.0, *p* < 0.001). The rate of mental health emergencies increased as well; however the *p*-value was high (*p* = 1.000).

The overall emergency rates differed during the lockdown “light” period (decline of −3.6% (SD 7.4, *p* < 0.001)) (Table 2). Stratified analyses looking at the dispatch keyword categories showed that, even though less pronounced than during first lockdown, the strongest declines were observed for pediatric emergencies (−16.7% (SD 24.8, *p* < 0.001)) and traffic accidents (−14.7% (SD 26.1, *p* < 0.001)). Compared to the two previous years, mental health emergencies increased during the lockdown “light” period (+34.3% (SD 65.5, *p* < 0.001)). The rates were also statistically significantly increased in the dispatch keyword categories “pain” and “respiratory problems”.

During the first weeks of the second lockdown, the overall emergency rate declined by 7.3% (SD 8.5), *p* < 0.001) (Table 3). A decline of 9.3% (SD 10.4, *p* < 0.001) was observed in sparsely populated rural districts, similar to the decline in large cities (−8.1% (SD 7.6, *p* = 0.004). The most frequent reason for dispatch was cardiovascular complaints (2018/2019: 115.8 (SD 22.8), 2020: 109.1 (SD19.4), which declined by 4.6% (SD 13.3, *p* = 0.002). During this period, emergency rates were lower than in 2020 in seven out of ten dispatch keyword categories, with the largest reductions in the rates for traffic accidents, pediatric emergencies and trauma. The rates of mental health emergencies and respiratory problems were statistically significantly higher than in previous years.

During the summer holidays, there was no statistically significant difference in the emergency rate overall (*p* = 0.225) nor in many sub-categories (see Table 4). For three out of ten dispatch keyword categories (trauma, other emergency and consciousness) the emergency rates were reduced compared to the previous years.

Figure 2 displays the distribution of the relative change in transport rates for each defined period. During the first lockdown, the median difference was −9.0% (first quartile −11.6, third quartile −6.7). During the lockdown “light” period, the relative change was −4.7% (−6.6 to−1.3), −7.0% (−9.6 to−3.7) during the first weeks of the second lockdown and −2.8% (−4.7 to−1.3) during the Bavarian holidays. 

## 4. Discussion

Our retrospective observational study based on the Bavarian data showed a 2.6% reduction in the yearly number of emergencies per 100,000 population in 2020, compared to the year 2018, and a 4.8% reduction compared to 2019. During all the periods, dispatch without pre-hospital emergency physicians was more frequent and the emergency rates were highest in large cities. The most frequent reasons for dispatch were cardiovascular problems, trauma and other emergencies; the rarest reasons for dispatch were mental health emergencies, pediatric emergencies and traffic accidents. A lower number of emergencies per 100,000 population was particularly observed during the periods with strict lockdown measures, with the sharpest declines occurring during the first lockdown. During the lockdown periods, the reduction was most pronounced in the large cities. Compared to the two previous years, the largest declines in the emergency rates during lockdown periods were observed if the dispatch keyword categories “traffic accident”, “pediatric” or “trauma” were assigned, whereas the largest increases were observed for the keyword categories “mental health” and “respiratory”. However, the biggest relative changes related to keyword categories that were less frequent compared to other categories. During the summer holiday period, where many restrictions were lifted, emergency rates were similar to the rates during the same period in 2018 and 2019. The transport rates were consistently below the level of previous years, even during the first weeks of the year 2020. The reduction in the transport rates was more pronounced at the start of the pandemic, especially during the first wave and the beginning of the second wave.

We observed lower emergency rates during the lockdown periods (12.9% reduction during the first and 7.3% reduction during the first weeks of the second lockdown), whereas the rates did not significantly differ during the summer holidays. A drop in the total number of rescue missions by 23.02% during the lockdown was observed in the city of Frankfurt [7], and a drop of 14.2% in the city of Leipzig [6], which matched our finding that the reduction in emergency rates was more pronounced in the large cities. Yet, data from the less densely populated German region of eastern Lower Saxony also showed similar declines [8]. Studies from other countries have also mostly reported fewer responses during periods with restrictions [13,14,15,16,17,18,19]. Yet, there are also reports of increasing numbers of patients accessing emergency medical services during the first weeks of the COVID-19 outbreaks in northern Italy [20] and Calgary, Canada [21]. This could be explained by increased healthcare needs [20] or patients calling EMS specifically to be assessed for COVID-19 symptoms [21].

Even though there are other reasons for physician dispatch, conditions that require the attendance of a physician are usually more severe. Our analysis did not show consistent differences in the declines of emergencies with and without physician support throughout the year. In a northern German region, there was no difference in the disease severity (GCS and NACA score) between the first lockdown and the previous weeks [8]. In New Zealand, Dicker observed that the proportion of lowest acuity increased [22]. Canadian data showed a mixed picture: there were reports of a greater proportion of moderate [21] and high acuity [17] calls, but also reports of fewer occurrence of critically ill patients [16] and unchanged severity [21].

Emergency rates in almost every keyword category declined. A marked decrease during the lockdown periods was observed for emergencies in the categories “traffic accident” and “trauma”. This is consistent with many other studies, that also observed a decrease in the activation of emergency medical services related to patient injury and trauma [6,7,15,22,23] and traffic accidents [13,22]. “Stay at home” restrictions might have been largely followed. Thus, with decreased mobility, the risk of injury during recreational activities or in traffic accidents decreased as well.

We also found a reduction in the emergency rates during lockdown for pediatric emergencies. Other authors have reported lower proportions of younger EMS patients [8,13,20]. Pediatric ED visits also fell sharply [24]. This could be due to the reduced exposure to injury and other infections, as schools and recreational facilities were closed. Parents also access emergency departments with their child for reassurance, convenience and easy access [25]. This could also be true for EMS and parents might have thought twice before calling pre-hospital EMS for their child, as they feared exposure to COVID-19 was associated with the utilization of these services at that time.

The emergency rates for conditions indicating respiratory problems was one of the few categories where the rates increased during all periods. An increase in the EMS activation during the first weeks of the pandemic also occurred in Leipzig, Germany [6], and British Columbia, Canada [16], whereas other studies found a reduction of respiratory diseases during the pandemic in spring 2020 [8,13,22]. On one hand, fewer respiratory illnesses seem plausible as the consequence of non-pharmaceutical interventions. Transmission of other notable infectious diseases in Germany were indeed less frequent than in previous years [26]. On the other hand, COVID-19 causes respiratory illness, and patients might have been more alert to these symptoms. Furthermore, even if the dispatch centers had other ways to flag calls with suspected COVID-19, in order to prepare crews to use protective equipment, it is possible that keywords indicating respiratory disease were used for suspected cases, even if patients were actually calling because of a different illness.

The other keyword category with increasing emergency rates during all periods was “mental health”. Other EMS studies also found that these conditions increased during lockdowns [7,13,16,22,23]. Andrews et al. found an initial reduction in mental health related presentations at the beginning of lockdown, which increased over the course of lockdowns and decreased after the restrictions were relaxed [13]. We also observed an increase in the rates of mental health emergencies during the Bavarian summer holidays, a period with fewer restrictions. An increase in this category was already identified at the beginning of the year, when most people were still unaware of the risk of the pandemic. The rising utilization of EMS for mental health problems in 2020 may, therefore, reflect a general trend reinforced through the pandemic. It is, therefore, a hopeful sign that experts agree that the pandemic could be an impetus to improve mental health care for everyone [27].

A lower proportion of patients transported to a hospital is also consistent with many other studies [6,13,14,21,22,23]. Lane found that mainly the categories of lower-priority patients were less frequently transported to a hospital [21]. This suggests that, if the conditions was not severe, EMS personnel might have made the decision to leave a patient at home out of a fear of exposure to COVID-19 at a healthcare facility. It might also have been that patients themselves were more reluctant to be transported for the same reason. It might also have been that personnel were trying to avoid unnecessary transports, in order to not strain emergency departments and hospitals already operating at high capacity.

The comparison of the use of pre-hospital EMS to the use of emergency departments during the early pandemic periods show some parallels: The most pronounced decreases occurred during the first and second wave [28]. Also, the decrease was stronger in young age groups [28,29] and in patients affected by trauma [6,30]. Other results were not on the same line: Studies looking at emergency department use report declines in patients with neurological symptoms [6,31,32] and cardiovascular disease [32]. Cardiovascular complaints were the most frequent reason for EMS use, yet we did not detect pronounced declines in these categories, except during the first lockdown. Overall, there were different reasons for changing utilization during the pandemic. On the one hand, the burden of disease increased due to COVID-19 patients. Yet, the indirect effects of COVID-19, like the hesitance to use medical services out of fear of infection, postponed consultations, and difficulty in seeking medical advice due to lockdowns and movement restrictions also affect the disease burden [33]. There is evidence that the hospital admission decreased during outbreaks even for acute medical conditions: reductions in myocardial infarction hospital admissions were observed [34,35,36] as well as a decreasing numbers of patients with stroke [35,36,37,38]. Yet the incidence of out-of-hospital cardiac arrests and cardiac arrest deaths increased during the COVID-19 outbreaks in Italy [34,39] and Paris, France [33]. Campo et al. suspect that some patients ignored the myocardial infarction symptoms too long, and that a part of the out-of-hospital cardiac deaths might be the “missing” myocardial infarctions [34]. In Bavaria, the indirect effects seem to have outweighed the number of additional COVID-19 cases requiring an ambulance: during most weeks of the year and especially in periods with a relatively high incidence and with lockdown measures, the emergency rates in 2020 declined compared to previous years. Unfortunately, our analysis was not able to show whether mildly or severely ill patients drove the reduction in emergency rates. Patients might have avoided using pre-hospital EMS for both serious and less urgent conditions. It is also possible that staying at home reduced the likelihood of being exposed to illness and injury, and, thus, led to a reduced number of emergencies. 

### Limitations

Our analyses relied on routinely collected data. The dispatch records were created automatically if a vehicle was dispatched and we, therefore, believed a documented dispatch to be a reliable measure for the activation of an ambulance. Information about location and keyword are important for tactical reasons and to inform ambulance crews and we, therefore, believed this information to be reliable most of the time. The same was true for the time stamps and the destination of transport. However, missing time stamps might occur and could lead to an underestimation of the transport rates. Dispatch in Bavaria is keyword-based and the assignment of keywords might vary between centers and over time.

In contrast to calendar weeks, the defined periods did not always comprise of exactly the same weekdays. As emergency rates differed between the weekdays and weekends, this might have influenced the comparison between the periods. The Bavarian summer holidays in the two previous years started two and three days later, respectively. However, we believe that a few days did not severely bias the overall trend during those periods. In addition, when comparing different periods, it has to be kept in mind that some comprise fewer days than others do. In addition, even though similar restrictions were in place during the different lockdown periods, the compliance with COVID-19 policies and their enforcement likely changed over the course of the pandemic. Comparison to other regions was difficult, especially if the population composition, healthcare infrastructure and COVID-19-policies were very different. Our findings may, therefore, only be relevant to Bavaria and may not be generalizable to other regions. 

## 5. Conclusions 

While there were concerns of an overwhelmed emergency medical services system, the opposite was the case in Bavaria in 2020. Compared to the two previous years, utilization declined. This showed that the ability to reorganize services and adjust capacities was important in order to react to the special circumstances. Matching health care needs and emergency medical resources remains a challenge at both ends of the spectrum: whereas an increase of unnecessary care to a pre-pandemic level would be unwanted, patients in need of immediate medical care must be encouraged to seek urgent care, even during a pandemic. More information on the disease severity and patient outcomes would facilitate the assessment of changing utilization in the future.

## Figures and Tables

**Figure 1 healthcare-11-01983-f001:**
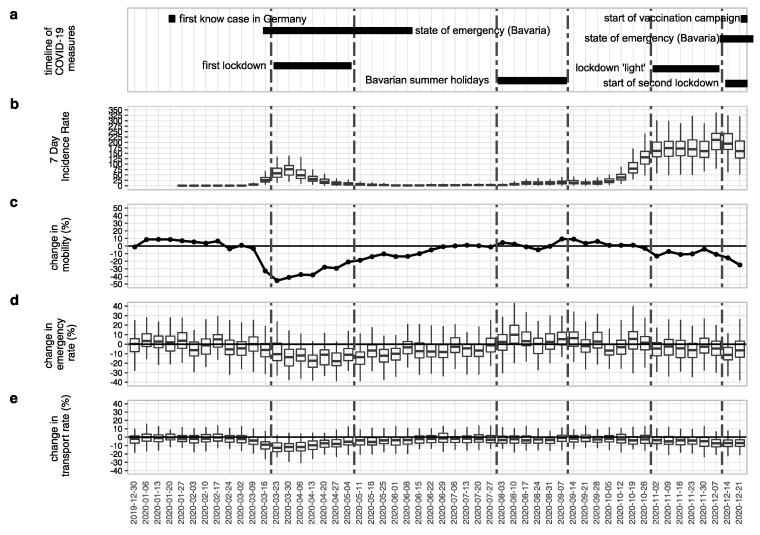
(**a**) Timeline of COVID-19 response measures; (**b**) Distribution of 7-day incidence across the 96 Bavarian districts during calendar weeks 1-52; (**c**) Percentage change in mobility in Bavaria during calendar weeks 1-52, compared to 2019, horizontal line represents no change; (**d**) Distribution of percentage change in emergency rates across the 96 Bavarian districts in 2020 compared to the average of both previous years during calendar weeks 1-52, horizontal line represents no change; (**e**) Distribution of proportion of transported patients across the 96 Bavarian districts, horizontal line represents median transport rate of the two previous years.

**Figure 2 healthcare-11-01983-f002:**
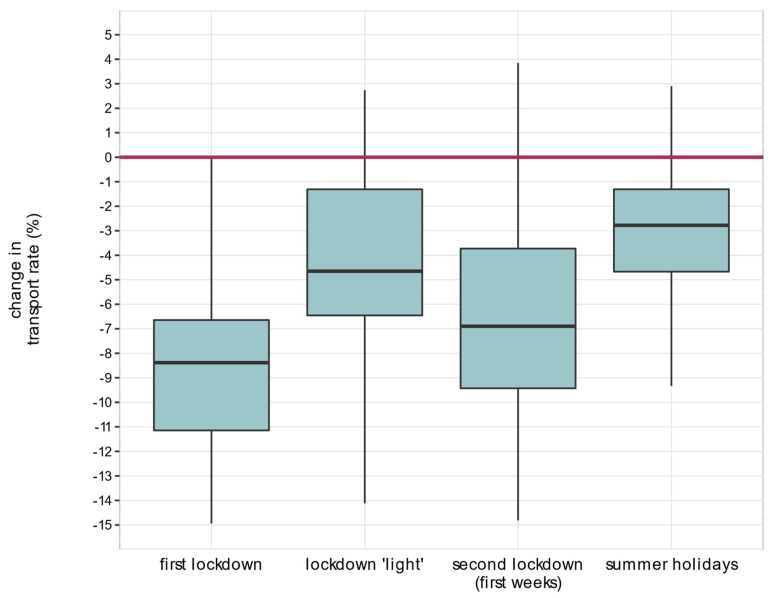
Boxplot of distribution of the percentage change in transport rates across the 96 Bavarian districts, by time period.

**Table 1 healthcare-11-01983-t001:** First lockdown (20 March–10 May).

	Emergency Rate Mean (SD)	Emergency Rate Mean (SD)	*p*	Percentage Change Mean (SD)
	First lockdown (2020)	Pre-pandemic (2018/2019)		
**overall**	963.6 (187.8)	1114.0 (237.1)	0.000	−12.9 (6.8)
**response**				
pre-hospital emergency physician support	410.9 (67.3)	475.5 (79.3)	0.000	−13.2 (7.9)
without pre-hospital emergency physician	552.7 (150.5)	638.5 (187.6)	0.000	−12.4 (9.6)
**rurality**				
large city	763.1 (334.8)	928.6 (411.2)	0.000	−17.2 (6.6)
urban	624.4 (252.9)	691.8 (280.5)	0.000	−9.6 (6)
predominantly rural	650.4 (283.2)	756.9 (339.5)	0.000	−13.1 (8.4)
rural	617.5 (261.4)	715.8 (302.4)	0.000	−13.4 (8.8)
**keyword category**				
pain	96.4 (20.1)	99.6 (21.0)	0.027	−2.1 (14.5)
respiratory	100.2 (30.8)	91.1 (20.3)	0.000	10.1 (20.0)
consciousness	67.6 (16.2)	83.9 (25.1)	0.000	−17.2 (14.0)
cardiovascular	228.4 (40.7)	249.7 (48.1)	0.000	−7.8 (10.1)
paediatric	29.1 (8.0)	42.0 (10.3)	0.000	−29.2 (17.6)
neurologic	92.8 (18.0)	104.1 (19.2)	0.000	−9.8 (14.9)
mental health	11.6 (6.8)	11.4 (5.9)	1.000	6.4 (52.6)
other emergency	112.0 (42.3)	137.6 (55.0)	0.000	−16.3 (15.7)
trauma	145.4 (38.8)	190.8 (54.0)	0.000	−22.8 (10.7)
traffic accident	26.0 (6.7)	38.6 (9.1)	0.000	−30.4 (20.1)

**Table 2 healthcare-11-01983-t002:** Lockdown “light“ (1 November to 9 December).

	Emergency Rate Mean (SD)	Emergency Rate Mean (SD)	*p*	Percentage Change Mean (SD)
	2018/2019	2020		
response				
pre-hospital emergency physician support	343.5 (58.7)	326.1 (52.6)	0.000	−4.5 (9)
without pre-hospital emergency physician	446.2 (133.1)	428.5 (116.4)	0.000	−2.8 (9.7)
rurality				
Large city	676.9 (292.5)	621.9 (269.4)	0.000	−8.0 (3.6)
urban	488.8 (199.5)	473.5 (187.1)	0.052	−2.5 (6.5)
predominantly rural	535.9 (240.8)	513.5 (218.4)	0.020	−3.0 (10.2)
rural	504.7 (211.1)	483.1 (202.8)	0.001	−3.9 (9)
keyword category				
pain	74.4 (17.9)	80.4 (18.4)	0.000	9.8 (19.1)
respiratory	63.3 (14.8)	73.8 (16.4)	0.000	18.4 (21.1)
consciousness	61.9 (18.5)	54.0 (15.0)	0.000	−10.4 (18.3)
cardiovascular	184.6 (33.5)	176.3 (30.4)	0.000	−3.7 (11.3)
pediatric	28.5 (7.7)	23.1 (6.9)	0.000	−16.7 (24.8)
neurologic	76.0 (14.3)	75.6 (15.5)	0.759	0.5 (16.1)
mental health	7.8 (3.7)	9.6 (5.3)	0.000	34.3 (65.5)
other emergency	96.8 (36.6)	85.0 (28.8)	0.000	−9.5 (15.6)
trauma	126.8 (38.4)	113.9 (34.1)	0.000	−9.3 (12.2)
traffic accident	23.7 (7.1)	19.5 (6.5)	0.000	−14.7 (26.1)

**Table 3 healthcare-11-01983-t003:** First weeks of second lockdown (9 December to 31 December).

	Emergency Rate Mean (SD)	Emergency Rate Mean (SD)	*p*	Percentage Change Mean (SD)
	2018/2019	2020		
response				
pre-hospital emergency physician support	215.8 (37.4)	201.3 (34.1)	0.000	−5.9 (11.7)
without pre-hospital emergency physician	284.6 (85.8)	257.3 (67.7)	0.000	−8.2 (10.1)
rurality				
large city	410.0 (177.3)	373.5 (159.5)	0.001	−8.1 (7.6)
urban	309.0 (125.6)	296.6 (116.6)	0.032	−3.3 (7.2)
predominantly rural	340.7 (153.9)	311.8 (132.3)	0.000	−6.9 (11.4)
rural	323.5 (140.6)	289.8 (118.7)	0.000	−9.3 (10.4)
keyword category				
pain	48.1 (11.9)	49.2 (10.7)	0.270	4.8 (20.5)
respiratory	43.6 (11.1)	51.4 (15.1)	0.000	19.7 (26.5)
consciousness	39.9 (11.4)	34.5 (9.0)	0.000	−10.3 (22.5)
cardiovascular	115.8 (22.8)	109.1 (19.4)	0.000	−4.6 (13.3)
pediatric	17.1 (4.7)	12.5 (4.3)	0.000	−23.3 (31.6)
neurologic	47.4 (9.0)	43.7 (8.0)	0.000	−6.5 (15.8)
mental health	5.2 (2.6)	6.0 (3.5)	0.022	34.6 (96.1)
other emergency	61.8 (22.7)	51.5 (16.7)	0.000	−13.8 (16.7)
trauma	78.8 (29.9)	66.2 (21.6)	0.000	−14.0 (17.1)
traffic accident	13.7 (4.3)	8.7 (3.4)	0.000	−32.7 (28.1)

**Table 4 healthcare-11-01983-t004:** Bavarian summer holidays (27 July to 7 September).

	Emergency Rate Mean (SD)	Emergency Rate Mean (SD)	*p*	Percentage Change Mean (SD)
	2018/2019	2020		
response				
pre-hospital emergency physician support	404.1 (69.6)	411.1 (68.1)	0.035	2.2 (8)
without pre-hospital emergency physician	546.8 (153.6)	548.5 (141.2)	0.765	1.4 (9.2)
rurality				
large city	765.6 (334.7)	785.3 (341.0)	0.144	3.2 (6.1)
urban	576.9 (239.0)	599.8 (240.9)	0.015	4.7 (6.4)
predominantly rural	636.2 (283.2)	636.9 (267.1)	0.930	1.4 (8.9)
rural	634.3 (265.2)	632.0 (267.5)	0.804	−0.1 (8)
keyword category				
pain	86.4 (19.2)	97.4 (21.9)	0.000	14.0 (16.9)
respiratory	69.1 (15.4)	72.7 (16.5)	0.003	6.6 (17.7)
consciousness	70.7 (20.5)	65.8 (18.8)	0.000	−5.4 (15.5)
cardiovascular	211.8 (38.7)	216.8 (38.4)	0.049	3.1 (10.9)
pediatric	30.1 (7.2)	31.1 (8.8)	0.160	4.6 (25.1)
neurologic	82.0 (15.4)	85.0 (19.6)	0.039	4.1 (16.5)
mental health	9.2 (4.4)	11.9 (6.6)	0.000	34.7 (61.7)
other emergency	125.5 (46.6)	118.6 (38.7)	0.022	−2.0 (20.4)
trauma	165.6 (45.6)	158.6 (44.9)	0.001	−3.5 (11.0)
traffic accident	38.5 (9.1)	39.5 (12.1)	0.265	3.7 (22.1)

## Data Availability

The data used in this paper was a combination of dispatch data, population data, spatial data, mobility data and reported COVID-19 incidence. Dispatch data are third-party data analyzed with permission of the Bavarian State Ministry of the Interior, Sport and Integration and the Bavarian social insurance agencies that the authors do not have the permission to distribute. Population data are publicly available from the Bavarian State Office for Statistics. Spatial data are publicly available from the German Federal Institute for Research on Building, Urban Affairs and Spatial Development. Aggregated mobility data are available from the Federal Statistical Office. Data of confirmed COVID-19 cases are available from the data platform of the Robert Koch Institute.

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
