# Peer review of "Pre-Hospital Emergency Medical Services Utilization Amid COVID-19 in 2020: Descriptive Study Based on Routinely Collected Dispatch Data in Bavaria, Germany"

_healthcare, 2023, doi:10.3390/healthcare11141983_

Round 1

Reviewer 1 Report

Dear editor, thank you for the opportunity to review the research article titled “Pre-Hospital Emergency Medical Services Utilization Amid COVID-19 in 2020: Descriptive study based on routinely collected dispatch data in Bavaria, Germany”. The aim of this study was to investigate the use of pre-hospital EMS during the COVID-19 pandemic throughout the entire year of 2020 on a superregional level, and to draw comparisons with previous years. The authors have used routinely collected dispatch data to conduct a retrospective descriptive study.

The study is well conducted, and its method is well described. Furthermore, the paper is well written and adds valuable data to the body of knowledge. Given the retrospective nature of this descriptive study it does come with several limitations that the authors describe and discuss.

Author Response

We thank the reviewer for taking the time to review our paper. We appreciated the kind comments and are happy to hear that you found our study to be valuable and insightful.

Reviewer 2 Report

Thank you for the opportunity to review this interesting manuscript. I would like to give you some suggestions.

Abstract:

I think the sentence “To determine the effect on utilization of pre-hospital emergency medical care, we investigate utilization throughout the year of 2020” should not be here (Lines 12-13)

Could you add other information in the background?

Results

I did not find the information referring to lines 176-177 in table 1.

I did not find the information referring to lines 188-189 in table 2.

I did not find the information referring to lines 197-198 in table 3.

Please, line 212:  Figure 2 (not figure 3)

Discussion

The 1st paragraph could include the summary of the study, for example: “Considering this retrospective study based on data from Bavaria, Germany, we observed a 2,6% reduction…

Limitations: The authors used data from Bavaria only.

Conclusion

The authors could briefly describe the results of their study.

Author Response

Comments and Suggestions for Authors:

Thank you for the opportunity to review this interesting manuscript. I would like to give you some suggestions.

Authors's response: we appreciate you taking the time to review are manuscript and are happy to here your suggestions.

Comments and Suggestions for Authors: Abstract:

I think the sentence “To determine the effect on utilization of pre-hospital emergency medical care, we investigate utilization throughout the year of 2020” should not be here (Lines 12-13)

Could you add other information in the background?

Author’s response (Abstract): We agree that this sentence does not provide background to the subject matter, but rather describes what we investigate. We changed the sentence and now put more emphasis on the potential avoidance of care in this context.

Comments and Suggestions for Authors: Results

I did not find the information referring to lines 176-177 in table 1.

I did not find the information referring to lines 188-189 in table 2.

I did not find the information referring to lines 197-198 in table 3.

Author’s response (results): The overall rate is reported in the results section yet not shown in the tables. The tables we submitted did include this information (first row). We will therefore kindly asked the editorial team to check why they are not displayed in the manuscript.

Comments and Suggestions for Authors: Please, line 212:  Figure 2 (not figure 3)

Author’s response: We apologize for the wrong label of the order of figures. We changed the annotation in the document.

Comments and Suggestions for Authors: Discussion

The 1st paragraph could include the summary of the study, for example: “Considering this retrospective study based on data from Bavaria, Germany, we observed a 2,6% reduction…

Author’s response (discussion): We thank the reviewer for this helpful comment. We added some context about the study to the first sentence of the paragraph.

Comments and Suggestions for Authors: Limitations: The authors used data from Bavaria only.

Author’s response (limitations): Unfortunately data collection of dispatch data is not standardized in Germany and regulated by federal states. The data is not available in every state and we did not acquire data from other federal states. We added a sentence to the limitations that our findings might be generalizable.

Comments and Suggestions for Authors: Conclusion

The authors could briefly describe the results of their study.

Authors's response (conclusion): We added a sentence to the conclusion that points out that utilisation declined, which we consider the main result of our analyses.

Reviewer 3 Report

Hegenberg et al. did a very thorough, detailed and extensive study about utilization of pre-hospital emergency services in Bavaria during the pandemic, comparing the data to non-pandemic times. They found that utilization of emergency services was decreased during the lockdown. This is result is not surprising result but nevertheless worthwhile to document. The manuscript extensive, written very clearly, conclusions are backed up by the results, limitations are mentioned adequately. I have no major comments and only a few minor comments.

Line 61: two-tired should be two-tiered

Line 141: what is the red horizontal line? I only see black lines

there are a few typos

Author Response

Comments and Suggestions for Authors

Hegenberg et al. did a very thorough, detailed and extensive study about utilization of pre-hospital emergency services in Bavaria during the pandemic, comparing the data to non-pandemic times. They found that utilization of emergency services was decreased during the lockdown. This is result is not surprising result but nevertheless worthwhile to document. The manuscript extensive, written very clearly, conclusions are backed up by the results, limitations are mentioned adequately. I have no major comments and only a few minor comments.

Authors's response: We thank your for your kind comments and are happy to address the minor issues raised.

Line 61: two-tired should be two-tiered

Author’s response: Thank you for pointing out this typo, which we corrected.

Comments and Suggestions for Authors: Line 141: what is the red horizontal line? I only see black lines

Author’s response: Thank you for pointing out that there are no red lines. A previous, colored version of the graph comprised a red horizontal line. We decided on black and white graphs for submission yet did not change the annotation. We now corrected this in the document.

Comments and Suggestions for Authors: Comments on the Quality of English Language

there are a few typos

Author’s response: We went over the manuscript to detect and correct additional typos.

Reviewer 4 Report

Authors described the trend of utilization of EMS during the early phase of the COVID-19 pandemic. The authors precisely evaluated weekly data associated with EMS and COVID-19 incidence during the whole study period. In general, the manuscript is well-written, includes enough data, and keeps consistency with previous relevant data.

Minor concerns

1. There are numerous comparisons in the present study. However, no corrections for the comparisons were done.

2. The heading of Table 1-4 are not easy to understand. Reviewer thinks that the heading should be “First lockdown (2020)” and “Pre-pandemic (2018/2019)”. In addition, the columns in Table 1 would be better if they were reversed. In other words, the left side should represent "First lockdown (2020)," and the right side should represent "pre-pandemic (2018/2019)."

Author Response

Thank you for reviewing our manuscript and sharing your minor concerns. Please find our response below: 

Minor concerns

  1. There are numerous comparisons in the present study. However, no corrections for the comparisons were done.

Author’s response to 1.: We agree that an increasing probability of obtaining false positive results can be a problem when there are numerous comparisons. We stratify our data and show many comparisons for subgroups. However, we always compare the rate of year 2020 to the average rate of 2018 and 2019, and not different strata within a group. Thus we never compare the means of more than two related groups simultaneously. In this case, we believe that we do not need a correction for multiple comparisons.

  1. The heading of Table 1-4 are not easy to understand. Reviewer thinks that the heading should be “First lockdown (2020)” and “Pre-pandemic (2018/2019)”. In addition, the columns in Table 1 would be better if they were reversed. In other words, the left side should represent "First lockdown (2020)," and the right side should represent "pre-pandemic (2018/2019)."

Authos’s response to 2: We added new headings and hope that this helps readers to understand the tables more easily. As suggested, we also switched the columns so that the year 2020 is the first column.